# Variation in the Composition and Quality of *Nigella sativa* L. Seed Oils—The Underestimated Impact on Possible Health-Promoting Properties

**DOI:** 10.3390/molecules29061360

**Published:** 2024-03-19

**Authors:** Grzegorz Dąbrowski, Sylwester Czaplicki, Iwona Konopka

**Affiliations:** Chair of Plant Food Chemistry and Processing, Faculty of Food Sciences, University of Warmia and Mazury in Olsztyn, Pl. Cieszyński 1, 10-726 Olsztyn, Poland; sylwester.czaplicki@uwm.edu.pl (S.C.); iwona.konopka@uwm.edu.pl (I.K.)

**Keywords:** *Nigella sativa* L., black cumin, volatile compounds, thymoquinone, tocols, quality indices

## Abstract

*Nigella sativa* L. (black cumin) is one of the most investigated medicinal plants in recent years. Volatile compounds like thymoquinone and unsaponifiable lipid compounds are crucial functional components of this oil. Unfortunately, the composition of oils and their quality indicators are ambiguous both in terms of identified compounds and value ranges. Thirteen oils were extracted with hexane from black cumin seeds grown in India, Syria, Egypt, and Poland and analyzed for their fatty acid composition, unsaponifiable compound content and volatile compounds. Oils were also subjected to quality tests according to standard methods. The fatty acid composition and sterol content/composition were relatively stable among the tested oils. Tocol content varied in the range of 140–631 mg/kg, and among them, β-tocotrienol and γ-tocopherol prevailed. Oils’ volatile compounds were dominated by seven terpenes (p-cymene, α-thujene, α-pinene, β-pinene, thymoquinone, γ-terpinene, and sabinene). The highest contents of these volatiles were determined in samples from Poland and in two of six samples from India. High acid and peroxide values were typical features of *N. sativa* L. oils. To sum up, future research on the medicinal properties of black cumin oil should always be combined with the analysis of its chemical composition.

## 1. Introduction

Seeds of *Nigella sativa* L. are abundant in oil (21–45%), carbohydrates (ca. 25%), protein (ca. 21%), dietary fiber (ca. 6%), and low-molecular bioactive compounds [1,2,3]. This plant is called black cumin, black seed, cumin noir, or black caraway in English [4]. Its seeds are currently used in a variety of food preparations, such as bread, cookies, cheese, yoghurt, meatballs, mayonnaise, ice cream, pork pies, and ground mutton, and for the production of valuable oil [5,6]. Simultaneously, *N. sativa* L. is one of the most investigated medicinal plants in recent years. Cumulative trends in PubMed and Medline databases showed ca. 10- and 20-fold increases in publications on *N. sativa* L. from the beginning of the twenty-first century up to 2020, respectively [4]. Similarly, in the last two years, the number of publications on “*Nigella*” deposited in PubMed was ca. 280 per year [7]. This significant scientific interest is based on the plenty of documented positive impacts on human health after the intake of seeds or their preparations [8]. This positive recommendation results from the long-term history of usage, e.g., Ayurveda medicine [9], and many pharmacological studies conducted in last few years. Currently, in the online database “www.clinicaltrials.gov”, there are 31 clinical studies on *N. sativa* L. with the status “completed” (25 studies) and the status “recruiting” (six studies)—access date: 25 July 2023. These clinical and other animal-based studies confirmed the analgesic; anticancer; antidiabetic; anti-inflammatory; antimicrobial; antioxidant; antitussive; broncho-dilating; gastro-, hepato-, and neuroprotective; immunomodulatory; and spasmolytic effects of the *N. sativa* L. oil [3,6,10,11].

Unfortunately, the majority of these studies utilized poorly defined *N. sativa* L. preparations (oils, seeds, seed extracts, or ointments). For example, in the study entitled “The benefits of *Nigella sativa* oil supplementation on asthma inflammation: A randomized, double-blind, placebo-controlled, phase II trial”, the registered Marnys^®^ black cumin oil was used [12], but its composition/quality was not confirmed. In another clinical study entitled “Effects of *Nigella sativa* oil on pain intensity and physical functions in patients with knee osteoarthritis: a randomized controlled trial” [13], “1 mL of *Nigella sativa* oil applied for two minutes, three times a day for 21 days” was tested, without detailed characterization of the oil used. Similarly, in a completed clinical study entitled “The clinical assessment of *Nigella sativa* oil vs. chlorohexidine as a therapeutic aid for gingivitis, effect on gingival il-6 and il-18 and antimicrobial efficacy”, the results were based on the use of “Group 1: Maintained adequate plaque control levels using mechanical methods + *N. sativa* oil (5 mL oil + 5 mL water) pulling for 3 min twice daily in the morning and at night”. These few examples show that clinicians treat *N. sativa* L. oil as a stable/constant preparation, but chemists and food science specialists underline the high non-uniformity of and variability in the composition and quality of *N. sativa* L. oils. These declarations are based on a detailed analysis of recently published manuscripts that show the extremely variable composition and quality of *N. sativa* L. seeds and oils on the worldwide market. For example, they differ in their contents of thymoquinone (TQ), a component of essential oils recognized as a key health-promoting ingredient [14,15]. TQ content in the commercial black cumin preparations (oils, softgel or hard capsules) in Malaysia ranged from 0.07% to 1.88% wt/wt [16]. Similarly, TQ contents in six black seed oils and five black seed oil-containing capsules originating primarily from Egypt varied from 3.08 to 809.4 mg/100 g [17]. Determined TQ share in volatiles varied from ca. 2% in seeds cultivated in India to ca. 100% in seeds cultivated in Syria [14]. Similarly, also variable are fatty acids and unsaponifiable lipid components [18,19], as well as oil quality indices: peroxide value, induction time, acid value, etc. [6,18,19]. For example, the peroxide values of black cumin oils can be very low (ca. 0.2 meq O_2_/kg—Mazaheri et al. [18]) or extremely high (ca. 124 meq O_2_/kg—Szydłowska-Czerniak et al. [19]).

The variation in black cumin seeds/oils depends mostly on the seed’s origin and method of oil extraction [3,18]. Black cumin is an annual herbaceous plant, and it is native to warmer areas of middle Europe (e.g., Romania, Italy) and continues to Eurasia and the Levant. Currently, it has also been transplanted to an area stretching from Morocco to Pakistan and India to China and Malaysia [5]. Since the vegetation period required to obtain mature seeds is ca. 100–120 days [20], black cumin plantations are also found in colder regions, such as Germany [21]. The majority of the global commercial supply of these seeds is obtained from cultivation in India [22]. The estimated early world production is ca. 600 thousand tons, with an 86% share in India, followed by Iran (4%), Syria (3%), and Turkey (2%). The remaining 5% comes from different countries such as Egypt, Pakistan, and Afghanistan [23]. In contrast, the top 10 countries that have the highest export values of black cumin oils are, in order, Malaysia, Indonesia, China, the Netherlands, Germany, Sweden, India, the United States, Russia, and Spain [24].

The method of *N. sativa* L. seed pre-treatment and oil obtainment is also significant for oil composition and quality. For example, Kiralan et al. [25] found that cold pressing, in contrast to Soxhlet and microwave-assisted extraction, resulted in the lowest induction period and the highest peroxide value, accompanied by the highest total phenolic, tocopherol, and TQ contents. In other studies, it was confirmed that dry air or infrared roasting positively affects the quality of this oil [26].

The presented introduction showed that although *N. sativa* L. oils have been examined by several scientific teams, their typical compositions and quality levels are still disputable. The main aim of the current study is to show variations in the compositions and quality levels of black cumin oils extracted from selected batches of seeds cultivated in Poland, India, Syria, and Egypt. The seeds differed in their expiration date (June 2023–January 2025), and oils were obtained via the same extraction method (Soxhlet apparatus using n-hexane as solvent).

## 2. Results and Discussion

### 2.1. Variation in Oil Yield and Fatty Acids Composition

The oil yield of the *N. sativa* L. seeds varied from 35.7% (sample No. 12/Poland) to 44.1% (sample No. 11/Egypt), with an average value of 39.6 ± 2.9% (Table 1). These values were compatible with those previously presented by Mazaheri et al. [18], Matthaus and Musa Özcan [21], and Prakhova [20]. The oils were composed of 14 fatty acids (Table 1, Appendix A), while the average main fatty acid shares were, in order, C18:2 (ca. 55.4%), C18:1 (ca. 24.8% in total; with ca. 1% of *trans* isomer), C16:0 (ca. 12.7%), C18:0 (ca. 3.1%), and C20:2 (ca. 2.1%), making the total share of these five fatty acids at least 98%. Among these major fatty acids, the highest variation was determined for stearic acid (CV = 12.1%), while the lowest was for linoleic acid (CV = 3.7%). On average, saturated fatty acids represented 16.3% (with a range from 14.6% to 17.3%), mono-unsaturated represented 25.9% (21.1–27.5%), and double unsaturated represented 57.5% (55.4–64.1%). The share of α-linolenic acid did not exceed 0.45%. Among the tested oils, one sample (No. 12) extracted from seeds cultivated in Poland was clearly distinguishable, having the highest share of linoleic acid and the lowest shares of saturated and mono-unsaturated acids.

The presented shares of fatty acid are close to the results obtained by other scientists [1,19,27,28]. This indicates that the fatty acid composition of *N. sativa* L. seeds, independent of cultivation area, is relatively constant. Different compositions found in some studies (e.g., approx. 23% α-linolenic acid, total polyunsaturated fatty acids approx. 15%, or ca 13% of *trans*-9-elaidic acid) are exceptional cases [2,19,29] and probably misleading samples.

### 2.2. Variations in Unsaponifiable Compounds

The total sterol contents in the 13 analyzed *N. sativa* L. oils varied from 1413 (sample No. 4 from India) to 2071/2074 (samples Nos. 12 and 2 from Poland and India, respectively), with an average value of 1893 ± 188 mg/kg (Table 2, Appendix A). β-sitosterol prevailed in all oils, reaching contrast values ranging from 537 to 959 mg/kg (samples Nos. 4 and 2 from India). The share of this compound changed from ca. 35% (samples Nos. 1 and 12 from Poland and India) to 47% (sample No. 9 from Syria), with an average value of ca. 41%. The next-most important sterol was cycloartenol, with a content ranging from 379 (sample No. 9/Syria) to 681 mg/kg (sample No. 12/Poland), with a ca. 25% share of total sterols, on average. Each of the following sterols accounted for at least 10% in all samples: stigmasterol, isofucosterol, and 24-methylene cycloartenol. The lowest content was determined for campesterol. This sterol was absent in one sample originating from Syria, while the highest content was determined in sample No. 10 from Egypt (131 mg/kg). The majority of samples (Nos. 11 from 13) contained small amounts of other non-identified sterols. Additionally, three oils contained squalene contents ranging from 12.6 to 24.6 mg/kg (samples Nos. 4 and 5 from India and No. 13 from Poland).

Data on sterols in black cumin oils in previous studies vary widely. Contents vary from 1661 [30] to 20,830 mg/kg of oil [31], showing ca. 12-fold possible variation. However, as in the current study, the majority of studies show the total content of sterols at the level of 1700–3000 mg/kg [19,21,30]. It should be noted that all studies confirm the dominance of β-sitosterol, with the share of this compound ranging from 32% to 64% of the total sterols. Other frequently determined sterols are Δ5-avenasterol, campesterol, and stigmasterol [19,32]. To date, squalene has hitherto rarely been identified in the seed oil of *Nigella sativa* [33]. However, in these seeds, the presence of squalene epoxidase (an enzyme that participates in the biosynthesis of saponins, phytosterols, etc.) was previously confirmed [34]. It is hypothesized that small amounts of squalene probably indicate unfinished metabolic pathways to these end products. In conclusion, both the contents and compositions of sterols seem to be relatively constant features of black cumin oils.

The content of tocols (Table 3, Appendix A) varied from 140.0 (sample No. 6/India) to 630.7 mg/kg (sample No. 12/Poland), with a high coefficient of variation (54.5%). In twelve oils, β-tocotrienol was the major compound, with its share ranging from 65.5 to 100% of total tocols. Surprisingly, this compound was found as the only tocol form in five oils (samples Nos. 6 and 8–11 from Egypt, India, and Syria). The content of γ-tocopherol prevailed (57.1%) in only one oil (No. 12) from seeds cultivated in Poland, while the second-highest β-tocotrienol share was 40.2%. Four oils also contained small amounts of α-tocopherol (samples Nos. 2, 7, 12, and 13 from Syria, India, and Poland), while δ-tocotrienol was found only in four oils from plants cultivated in India (Nos. 1, 3–5).

The literature data on tocol contents in *N. sativa* L. oils vary from ca. 20 [19] to 1731 mg/kg of oil [30], but the majority of studies show values close to the results found in the current study [18,21,35]. It should also be emphasized that β-tocotrienol, as the predominant or one of the main tocols present in black cumin oils, was previously found in two studies [21,30]. Other studies show that the main compound is γ-tocopherol, which can reach up to 86% of the total tocol content [18,19], and that the contents of α- and γ-tocopherols are comparable [32] or α-tocopherol is prevalent [35]. Thus, it appears that the topic of black cumin tocols is still open. Since α- and β-tocopherols/tocotrienols are the last compounds in the tocols biosynthetic pathway [36], further studies should examine if the variation in black cumin oils is the result of the seed ripening degree or identification errors (tocotrienol standards deficit).

In addition, it is worth of knowing that in Poland, the May–September photoperiod is much longer than in India, Syria, and Egypt. In general, long-day treatments, in contrast to short-day treatments, promote an increase in dry weight in plants [37]. We hypothesize that such a phenomenon may be related to the relatively high accumulation of bioactive compounds in seeds cultivated in Poland. We propose conducting studies with various photoperiods in black cumin cultivation.

### 2.3. Variations in Quality Indices

Selected quality indices of *N. sativa* L. oils are presented in Table 4. The acid values of freshly extracted oils in the study ranged from 19.7 (sample No. 5/India) to 51.6/52.3 mg KOH/g (samples Nos. 12 and 2 from Poland and India, respectively). It is known that these are very high values and were first reported in black cumin oils (it was confirmed that the reagents and solvents used were of perfect quality, using other oils and obtaining low acid values). Previously published data gave contrasting values to these values, ranging from 1.35 [26] to 21.86 mg KOH/g of oil [19].

The peroxide value is the next most frequently used indicator of oil quality. In the current study, these values ranged from 11.6 (sample No. 7 from Syria) to 71.6 meq O_2_/kg (sample No. 12 from Poland). Even higher variation was found previously since these values can range from 0.22 [18] to 123.8 meq O_2_/kg of oil [19]. According to the Codex Alimentarius Commission [38], virgin and cold-pressed oils should have up to 15 meq of O_2_/kg oil, while for other oils, this value is limited to 10 meq of O_2_/kg oil. It is worth noting that a lot of studies show peroxide values for *N. sativa* L. oils above 15 meq O_2_/kg (also in freshly extracted), larger than recommended in the mentioned Codex Alimentarius. Some studies suggest that these high peroxide values can be related to the essential oil content in these oils [39]. To explain this possible impact, we conducted an additional experiment by preparing sunflower oil (commercial sample with a fatty acid composition close to that of black cumin oil) supplemented by 0.25, 0.5, and 1.0% thymoquinone or p-cymene. The peroxide values of prepared oils were dose-dependent for the thymoquinone content addition, reaching corresponding values of 1.4 meq O_2_/kg in control oil and 12.5, 26.4, and 49.9 meq of O_2_/kg in supplemented oils, respectively. The second compound (p-cymene) did not affect the peroxide value of enriched oils. It points out that TQ may act as an oxidant of potassium iodide under conditions of peroxide value determination, but further examination of this activity can be conducted. Our prediction is based on the high Pearson correlation coefficient (r = 0.84) between the TQ content and the peroxide value in the tested oils.

It was also determined that samples with the highest peroxide values were additionally characterized by the higher percentage of conjugated dienes, whose maximal content was 0.22–0.25% in six samples from Poland and India. Surprisingly, ca. 10-fold higher contents of conjugated dienes (2.15–2.65%) at relatively lower peroxide values (4.07–6.11 meq O_2_/kg oil) were determined in a study conducted by Suri et al. [26].

### 2.4. Variation in Volatile Compounds of Oils

The results of the GC-MS analysis of the volatile compounds (Figure 1, Appendix A) indicate that the black cumin oils were mainly characterized by seven terpenes: p-cymene, α-thujene, α-pinene, β-pinene, thymoquinone, γ-terpinene, and sabinene. These compounds were detected in all the studied oils. Additionally, two unique compounds (α-phellandrene and α-terpinolen) were detected in two oils from plants cultivated in Poland and in sample No. 5 from plants cultivated in India. It is worth emphasizing that these three oils were characterized by the highest overall volatile compound content (Figure 1) and the highest number (20) of detected compounds. In other oils, from 16 to 18 volatile compounds were found. Particularly noteworthy is the fact that seed oils from Syria and Egypt were statistically significantly less rich in volatile compounds since the total volatile chromatographic response of these oils was more than 10-fold lower than that of oil No. 12 (Poland). Thymoquinone content was visibly the highest in four oils: two samples cultivated in India (Nos. 1 and 5) and both the samples cultivated in Poland. Typically, essential oils make up about 0.5–1.5% of *N. sativa* L. oil and are considered the main pro-healthy phytochemical of this oil [3]. Major active compounds have been isolated, identified, and reported so far in various black cumin seeds. The most important are thymoquinone, p-cymene, carvacrol, 4-terpineol, α-thujene, *trans*-anethol, longifolene, α-pinene and thymol [15]. However, previous studies suggested three chemotypes of the profile of essential oils (more details in Salehi et al. [3]). These include a “thymoquinone chemotype” in seeds from Egypt and Turkey, “*trans*-athenole chemotype” in seeds from Iran, and “p-cymene and thymol chemotype” in seeds from Morocco. All chemotypes were highlighted based on the major compounds in the studied oils. It appears that the thymoquinone chemotype could be the most pharmacologically active in black cumin oil [8]. Since this compound is the target product of the biosynthetic pathway from γ-terpinene, via p-cymene, and then via carvacrol and thymol to thymohydroquinone [40], the various *N. sativa* L. oils may suggest various ripeness degrees for seeds used for oil extraction.

### 2.5. Principal Component Analysis

PCA analysis of the studied *N. sativa* L. seed oils showed significant variations between these samples. Three separate groups can be distinguished based on the content of volatile components, oil composition, and oil quality (Figure 2). The first group is represented by only one sample, No. 12 (Poland), distinguished by the highest volatile compounds, tocols, and peroxide values. The second group includes samples Nos. 1, 3–5, and 13 (samples from India and Poland), recognized by their average contents of volatile compounds, tocols, and peroxide values. Finally, the third group are samples Nos. 2 and 6–11 (samples from India, Syria, and Egypt) with the lowest contents of volatile compounds, tocols, and peroxide values. The variation in samples with regard to volatile compounds was explained by PC1 with ca. 81% and PC2 with ca. 13% (Figure 2A), while the variations in the case of fatty acid shares, unsaponifiable compounds, and quality indices were explained by PC1 with ca. 43% and PC2 with ca. 23% (Figure 2B).

## 3. Materials and Methods

### 3.1. Plant Material

The material of the study consisted of 13 samples of *N. sativa* L. seeds originating from four countries (India, Egypt, Syria, and Poland) with expiration dates from June 2023 to January 2025. All samples were visually examined, and no significant differences in the colors, sizes, and shapes of the seeds were found between the examined samples. The moisture contents of these seeds were in range of 5.6–7.2%. Seeds were bought from a local market in Olsztyn (Poland) or via internet shopping in February 2023. A list of samples with the type of package and expiration date is presented below:India, glass jar (June 2023);India, paper and foil bag (February 2024);India, glass jar (March 2024);India, paper and foil bag (April 2024);India, cardboard box (November 2024);India, cardboard box (December 2024);Syria, foil bag (November 2023);Syria, foil bag (March 2024);Syria, foil bag (June 2024);Egypt, foil bag (March 2024);Egypt, foil bag (January 2025);Poland, paper and foil bag (November 2023);Poland, foil bag (December 2024).

### 3.2. Solvents and Reagents

Analytical-grade solvents and reagents such as anhydrous sodium sulphate, chloroform, methanol, sulphuric acid, diethyl ether, ethanol (99.9% purity), potassium hydroxide, and zinc powder were purchased from Chempur (Piekary Śląskie, Poland). Chromatographic solvents and reagents, such as silylating agents (pyridine (anhydrous, 99.8% purity) and *N*,*O*-bis(trimethylsilyl)trifluoroacetamide with 1% trimethylchlorosilane), n-hexane (≥95% purity, HPLC grade), and isopropanol (99.5% purity, HPLC grade), and standards, such as α-tocopherol (≥96% purity), γ-tocopherol (≥96% purity), δ-tocopherol (≥96% purity), p-cymene (≥99% purity) and thymoquinone (≥98% purity), and 5-α-cholestane (≥97% purity), were purchased from Sigma-Aldrich (Poznań, Poland). Helium for GC (99.999% purity) was purchased from Eurogaz-Bombi (Olsztyn, Poland). Deionized water was prepared with the use of the HLP 5 deionizer (Hydrolab, Gdańsk, Poland).

### 3.3. Experimental Methods

#### 3.3.1. Solvent Extraction

Before oil extraction, seeds were frozen in liquid nitrogen and carefully ground in an A20 laboratory mill (IKA-Werke, Staufen, Germany). A total of 25 g of ground seeds were weighed into a Soxhlet thimble and subjected to extraction via the Soxhlet method using 150 mL of n-hexane. After 18 h, the extraction was ended and the residual solvent was evaporated from the oil with the use of a R-210 rotary vacuum evaporator (Büchi, Flawil, Switzerland) at 45 °C. The oil content was determined by the weight of obtained oil in reference to the weight of each ground seed sample. The obtained oils were stored at −20 °C and used in all further determinations.

#### 3.3.2. Quality Indices

The acid and peroxide values were determined according to following procedures: ISO 660:2010 [41] and ISO 3960:2012 [42], respectively. The contents of conjugated dienes and trienes were determined according to the official AOCS Cd 7-58 method [43].

#### 3.3.3. Fatty Acids Composition

Before analysis, fatty acid methyl esters (FAMEs) were prepared according to the method described by Dąbrowski et al. [44]. A total of 0.02 g of each oil was weighed into glass ampule, and 2 mL of a chloroform/methanol/sulfuric acid (100:100:1, *v*/*v*/*v*) mixture was added. Ampules were hermetically closed and subjected to methylation (70 °C, 2 h). After that, ampules were opened, and zinc powder was added to neutralize the acid. Solvents were evaporated under nitrogen stream, and residual fatty acids methyl esters were dissolved in n-hexane (GC-MS purity). FAMEs were analyzed via the GC-MS technique using a GC-MS QP2010 PLUS (Shimadzu, Kyoto, Japan) apparatus in accordance with the parameters described by Dąbrowski et al. [44]. The column used for separation was a BPX70 (25 m × 0.22 mm × 0.25 μm) capillary column (SGE Analytical Science, Victoria, Australia), and the carrier gas was helium at a flow rate of 1.3 mL/min. The temperature of the column was programmed as follows: a subsequent increase from 150 °C to 180 °C at a rate of 10 °C/min, to 185 °C at a rate of 1.5 °C/min, to 250 °C at a rate of 30 °C/min, and then a 10-min hold. The GC-MS interface and ion source temperatures were set at 240 °C, and the electron energy was 70 eV. The total ion current (TIC) mode was used in the 50–500 *m*/*z* range.

#### 3.3.4. Phytosterols and Squalene Content

The same technique and apparatus used in the case of FAMEs were used for the analysis of phytosterols and squalene. Samples were prepared according to the method described by Dąbrowski et al. [44]. A total of 0.2 g of oil with 0.2 mL of 5α-cholestane internal standard solution (0.4 mg/g) was subjected to saponification and the extraction of unsaponifiable compounds. Obtained extracts were evaporated to dryness, re-dissolved in 1.5 mL of n-hexane, transferred into 1.5 mL chromatographic vials, and evaporated to dryness under a nitrogen stream. Next, 100 μL of pyridine and 100 μL BSTFA (*N*,*O*-bis (trimethylsilyl) trifluoroacetamide) with 1% TMCS (trimethylchlorosilane) were added to the dry residues and mixed. Obtained solutions were left at 60 °C for 60 min until silylation occurred. After derivatization, compounds were separated with the use of a ZB-5MSi (Phenomenex Inc., Torrance, CA, USA) capillary column with helium as a carrier gas at a flow rate of 0.9 mL/min. Moreover, 230 °C was the injector temperature. The following program of the column temperature was used: 70 °C for 2 min, a subsequent increase to 230 °C at a rate of 15 °C/min, to 310 °C at a rate of 3 °C/min, and then a 10 min hold. The GC-MS interface and ion source temperatures were 240 and 220 °C, respectively. The electron energy was set at 70 eV. The total ion current (TIC) mode for quantification (100–600 *m*/*z* range) was used. Compounds were identified based on their mass spectra compared with mass spectral libraries (NIST08 library, Shimadzu, Kyoto, Japan). Squalene and sterols were identified based on retention times and mass spectra, and their contents were calculated with reference to the internal standard and given as mg/100 g. The repeatability for 5-α-cholestane determination (expressed as a coefficient of variation) was 2.5%. The limit of quantification was 0.05 μg/g of oil.

#### 3.3.5. Tocols Content

The analysis of tocols was carried out using high-performance liquid chromatography coupled with fluorescence detection (HPLC-FLD), in accordance with the method described by Dąbrowski et al. [44]. Agilent Technologies (Palo Alto, CA, USA) 1200 series liquid chromatograph equipped with a fluorescence detector was used. Next, 1% (*m*/*v*) oil solutions in n-hexane (HPLC purity) were prepared and injected into the chromatographic system. Tocopherols were separated at a temperature of 25 °C with the use of a LiChrospher Si 60 column (250 mm × 4 mm, 5 μm) (Merck, Darmstadt, Germany). The mobile phase consisted of a 0.7% isopropanol solution in n-hexane at a 1 mL/min flow rate. The fluorescence detector was set at 296 nm for excitation and 330 nm for emission. Peaks were identified in comparison to retention times determined for α, β, γ, and δ tocopherol standards (Merck, Darmstadt, Germany) separately. Additionally, tocotrienol identification was conducted based on barley lipid sample analysis as a rich source of these compounds. The identification of tocols also resulted from a comparison with known NP-HPLC-FLD chromatograms of barley grain samples [45]. Tocol content was calculated using the external calibration curves of each tocopherol standard. The concentration of ethanolic stock standard solutions was evaluated spectrophotometrically with the use 75.8, 89.4, 91.4, and 87.3 as extinction coefficients for α-, β-, γ-, and δ-tocopherol, respectively. The calibration curves show good linearity (R2 ≥ 0.9967) for each of the tocopherols. The repeatability expressed as a coefficient of variation was not worse than 3.68%. The limits of quantification were, respectively, 0.45, 0.4, 0.4, and 0.2 μg/g of oil.

#### 3.3.6. Volatile Compounds Profiles

A total of 2 g of the sample was inserted into a 20 mL headspace screw top vial, and the vials were incubated at 40 °C for 300 s and shaken at 500 rpm. After incubation, the silicone septa was pierced by the syringe, and 2.5 mL of the headspace was injected into the gas chromatographic system. The analyses of volatile compounds were performed using an Agilent model 8890 series gas chromatograph in combination with a Gerstel MPS autosampler and an Agilent 7000D QQQ mass detector. The compounds were separated in a DB-624 (J&W Scientific, 30 m, 0.25 mm i.d., 1.4 μm film thickness), working with the following temperature program: 40 °C, followed by holding for 5 min; 11 °C/min up to 80 °C; and 22 °C/min up to 250 °C, followed by holding for 2 min. The temperatures for the injection port, ion source, quadrupole, and interface were set at 200, 230, 150, and 230 °C, respectively. Mass spectra were obtained in the electron impact at 70 eV in a scan range from *m*/*z* 10 to 200. The detected compounds were identified by comparing their mass spectra with the NIST 2017 MS library and published data [25].

#### 3.3.7. Statistical Analysis

Statistica 13.1 software (TIBCO, Palo Alto, CA, USA) was used for statistical analysis of the obtained results. The analysis of variance (ANOVA) with the Tukey’s test for homogenous groups was performed. Principal Component Analysis (PCA) was used for grouping *N. sativa* L. oils into distinguishable groups and to show correlation between the contents/shares of determined compounds and oil quality indices. All calculations were conducted at the *p* ≤ 0.05 significance level.

## 4. Conclusions

The results of the current study confirmed the high variation in the lipophilic phytochemical contents and quality indices of *N. sativa* L. oils extracted from 13 different samples of seeds originating from India, Syria, Egypt, and Poland available at a market. The lowest variation was determined for the fatty acid composition and sterol content composition. It was also shown that the main tocols in *N. sativa* L. oils are primarily β-tocotrienol and secondarily γ-tocopherol, which confirms the results of a few previous reports. The key accomplishment of this study was the finding of significant variations in *N. sativa* L. oils; in particular, there were no discernible links found with seed origin or the date of expiration. Additionally, this study highlighted unusually high levels of acid and peroxide values, which have seldom or never been reported previously. It suggests a high level of oil deterioration, but in the case of peroxide values, its high level can be caused by thymoquinone activity in this assay. PCA analysis showed three distinguishable groups of seed oils, characterized by various contents of volatile compounds, tocols, and quality indices. Surprisingly, oils from *N. sativa* L. seeds from plants cultivated in Poland were characterized by the highest/high levels of selected tocol, sterol, and volatile compound concentrations, although they were also accompanied by high acid and peroxide values. The expiration date declared on the *N. sativa* L. seed package had no significant effect on the composition or quality characteristics of the extracted oils.

In summary, we recommend that only black cumin oils with confirmed chemical composition, or at least with confirmed thymoquinone content, be used in future studies in human/animal/cellular models. Studies without this confirmation should be considered unreliable.

## Figures and Tables

**Figure 1 molecules-29-01360-f001:**
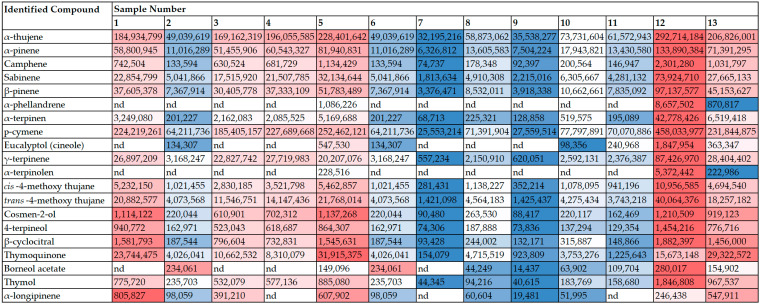
Results of headspace GC-MS analysis of volatile compounds of *N. sativa* L. seeds oils (values are expressed as peak area arbitrary units). Extremal data in each row are represented by various colors. The highest and the lowest values are marked by the most red and blue colors, respectively; nd—not detected.

**Figure 2 molecules-29-01360-f002:**
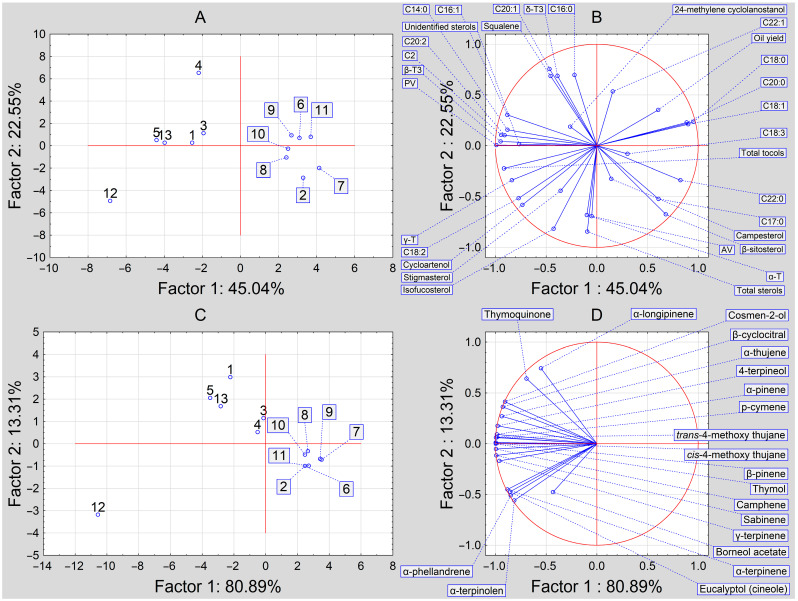
Projection of the cases (**A**,**C**) and the variables (**B**,**D**) onto the factor plane regarding the quality and composition (**A**,**B**) and volatile composition (**C**,**D**) of *N. sativa* L. seed oils.

**Table 1 molecules-29-01360-t001:** Oil recovery from *N. sativa* seeds and the fatty acid compositions of *N. sativa* L. seed oils.

Sample	Oil Yield (g/100 g)	Fatty Acids (% of Total Fatty Acids)												
		C14:0	C16:0	C16:1	C17:0	C18:0	C18:1	C18:2	C18:3	C20:0	C20:1	C20:2	C22:0	C22:1
1	40.51 ± 1.25 b	0.21 ± 0.00 cd	13.02 ± 0.09 de	0.19 ± 0.00 cde	0.07 ± 0.01 a	2.93 ± 0.04 cd	25.06 ± 0.07 de (1.08 ± 0.00 f)	55.32 ± 0.18 abc	0.14 ± 0.00 a	0.15 ± 0.03 ab	0.29 ± 0.01 cd	2.14 ± 0.02 c	0.03 ± 0.00 b	0.49 ± 0.08 ab
2	40.15 ± 0.90 b	0.17 ± 0.01 a	12.00 ± 0.01 ab	0.17 ± 0.01 abc	0.07 ± 0.00 a	3.30 ± 0.01 efg	26.37 ± 0.06 f (1.00 ± 0.01 bcde)	54.61 ± 0.01 ab	0.20 ± 0.01 abc	0.21 ± 0.00 bc	0.25 ± 0.00 abc	1.86 ± 0.02 ab	0.12 ± 0.01 d	0.71 ± 0.01 ab
3	39.50 ± 1.05 b	0.21 ± 0.01 cd	13.01 ± 0.04 de	0.19 ± 0.01 bcde	0.07 ± 0.00 a	3.06 ± 0.06 de	24.91 ± 0.07 d (0.93 ± 0.01 a)	55.15 ± 0.03 abc	0.15 ± 0.00 ab	0.16 ± 0.00 ab	0.30 ± 0.01 d	2.16 ± 0.00 cd	nd a	0.64 ± 0.14 ab
4	42.83 ± 0.07 bc	0.20 ± 0.01 bcd	13.05 ± 0.14 de	0.21 ± 0.00 e	0.06 ± 0.01 a	3.04 ± 0.01 d	24.29 ± 0.01 c (0.99 ± 0.03 bcd)	54.81 ± 0.25 ab	0.21 ± 0.01 bc	0.20 ± 0.03 abc	0.43 ± 0.01 e	2.32 ± 0.05 e	nd a	1.21 ± 0.07 ab
5	35.77 ± 1.23 a	0.22 ± 0.00 d	13.11 ± 0.03 e	0.20 ± 0.01 de	0.07 ± 0.01 a	2.65 ± 0.00 b	23.32 ± 0.01 b (1.01 ± 0.02 bcde)	57.01 ± 0.13 c	0.17 ± 0.01 ab	0.16 ± 0.02 ab	0.30 ± 0.01 d	2.30 ± 0.05 e	0.03 ± 0.00 b	0.50 ± 0.18 ab
6	41.84 ± 0.77 bc	0.20 ± 0.01 abcd	13.20 ± 0.01 e	0.16 ± 0.00 ab	0.08 ± 0.01 a	3.47 ± 0.01 gh	26.01 ± 0.04 f (1.05 ± 0.03 ef)	53.42 ± 0.33 a	0.16 ± 0.02 ab	0.21 ± 0.02 bc	0.26 ± 0.01 abcd	1.94 ± 0.03 ab	0.06 ± 0.00 c	0.86 ± 0.35 ab
7	41.62 ± 1.03 bc	0.17 ± 0.01 ab	11.77 ± 0.17 a	0.16 ± 0.01 ab	0.08 ± 0.01 a	3.47 ± 0.16 gh	26.25 ± 0.35 f (0.96 ± 0.01 ab)	55.07 ± 0.91 abc	0.16 ± 0.01 ab	0.21 ± 0.01 bc	0.25 ± 0.01 ab	1.83 ± 0.04 a	0.16 ± 0.01 e	0.44 ± 0.23 ab
8	40.49 ± 0.14 b	0.19 ± 0.01 abc	12.39 ± 0.04 bc	0.17 ± 0.01 abcd	0.08 ± 0.00 a	3.16 ± 0.1 def	25.33 ± 0.18 de (0.98 ± 0.01 abc)	55.41 ± 0.73 abc	0.44 ± 0.04 e	0.19 ± 0.02 abc	0.23 ± 0.01 a	1.83 ± 0.03 a	0.06 ± 0.00 c	0.57 ± 0.42 ab
9	35.91 ± 0.36 a	0.18 ± 0.01 ab	12.68 ± 0.15 cd	0.16 ± 0.00 ab	0.07 ± 0.00 a	3.40 ± 0.01 fgh	25.43 ± 0.17 e (0.98 ± 0.01 abc)	54.60 ± 0.45 ab	0.45 ± 0.00 e	0.20 ± 0.00 abc	0.25 ± 0.02 ab	1.89 ± 0.04 ab	0.08 ± 0.01 c	0.65 ± 0.52 ab
10	41.02 ± 1.25 bc	0.17 ± 0.01 a	12.66 ± 0.08 cd	0.15 ± 0.00 a	0.07 ± 0.00 a	3.33 ± 0.06 fg	25.05 ± 0.04 de (1.03 ± 0.01 cdef)	54.26 ± 0.88 ab	0.35 ± 0.02 d	0.20 ± 0.01 abc	0.25 ± 0.01 ab	1.97 ± 0.01 b	0.06 ± 0.02 bc	1.52 ± 0.74 b
11	44.11 ± 0.05 c	0.19 ± 0.01 abc	13.13 ± 0.08 e	0.16 ± 0.01 a	0.08 ± 0.01 a	3.59 ± 0.05 h	26.03 ± 0.05 f (1.04 ± 0.01 def)	53.54 ± 0.16 a	0.18 ± 0.01 ab	0.23 ± 0.00 c	0.27 ± 0.00 abcd	2.12 ± 0.04 c	0.08 ± 0.00 c	0.43 ± 0.10 ab
12	35.67 ± 0.02 a	0.22 ± 0.01 cd	11.93 ± 0.12 a	0.20 ± 0.01 de	0.07 ± 0.01 a	2.27 ± 0.02 a	20.59 ± 0.07 a (1.03 ± 0.01 cdef)	61.54 ± 0.16 d	0.24 ± 0.01 c	0.14 ± 0.00 a	0.27 ± 0.00 abcd	2.54 ± 0.03 f	nd a	nd a
13	35.96 ± 0.87 a	0.22 ± 0.01 cd	13.36 ± 0.16 e	0.20 ± 0.00 e	0.07 ± 0.01 a	2.80 ± 0.03 bc	23.20 ± 0.04 b (1.00 ± 0.01 bcde)	55.99 ± 0.93 bc	0.16 ± 0.01 ab	0.16 ± 0.01 ab	0.29 ± 0.01 bcd	2.29 ± 0.03 de	nd a	1.29 ± 0.75 ab
Mean	39.64	0.20	12.72	0.18	0.07	3.11	24.76 (1.00)	55.44	0.23	0.19	0.28	2.09	0.05	0.72
SD	2.90	0.02	0.53	0.02	0.01	0.38	1.61 (0.04)	2.06	0.11	0.03	0.05	0.23	0.05	0.41
CV (%)	7.31	9.88	4.19	11.40	8.29	12.14	6.50 (4.18)	3.72	47.24	15.28	17.92	10.80	95.32	57.58

a, b, …—Means in the same column for all variants followed by different letters are significantly different (*p* ≤ 0.05). Values are mean ± SD (*n* = 3); C18:1 is expressed as the total contents of *cis* and *trans* isomers (upper value) and the separate contents of *trans* isomers (lower value); SD—standard deviation; CV—coefficient of variance; nd—not detected.

**Table 2 molecules-29-01360-t002:** Phytosterol and squalene contents (mg/kg) in *N. sativa* L. seed oils.

Sample	Squalene	Campesterol	Stigmasterol	β-Sitosterol	Isofucosterol	Cycloartenol	24-Methylene Cyclolanostanol	Others	Total
1	nd a	87.88 ± 3.30 cd	193.28 ± 1.57 de	713.80 ± 12.54 bc	229.28 ± 9.67 de	540.21 ± 18.26 f	215.20 ± 1.69 f	29.99 ± 2.89 de	2009.63 ± 49.92 cde
2	nd a	113.36 ± 3.00 ef	172.16 ± 6.30 bcde	959.30 ± 12.91 g	210.68 ± 5.73 bcd	447.99 ± 1.46 bcd	155.98 ± 5.39 bc	14.56 ± 0.83 b	2074.03 ± 33.96 e
3	nd a	90.17 ± 1.79 cd	173.53 ± 3.92 bcde	688.78 ± 3.97 b	196.48 ± 2.71 bcd	506.10 ± 12.68 ef	193.79 ± 6.08 ef	23.45 ± 1.64 cd	1872.31 ± 27.38 c
4	24.65 ± 11.20 b	63.81 ± 0.91 b	115.63 ± 5.40 a	537.05 ± 10.26 a	144.31 ± 6.24 a	387.88 ± 3.08 a	150.29 ± 0.27 abc	14.44 ± 1.49 b	1413.41 ± 16.70 a
5	12.63 ± 2.12 b	89.58 ± 6.10 cd	191.31 ± 15.16 de	717.45 ± 20.44 bc	202.90 ± 6.87 bcd	526.67 ± 20.74 ef	210.68 ± 16.27 f	32.28 ± 4.14 e	1970.87 ± 89.73 cde
6	nd a	102.31 ± 5.23 de	144.09 ± 20.25 ab	832.26 ± 34.17 de	177.39 ± 22.15 abc	407.64 ± 11.98 ab	151.08 ± 0.45 abc	14.70 ± 2.02 b	1829.46 ± 96.24 c
7	nd a	105.17 ± 8.58 de	154.68 ± 7.28 bc	912.48 ± 20.24 fg	205.66 ± 4.14 bcd	481.60 ± 21.45 cde	164.41 ± 2.07 bcd	nd a	2024.00 ± 63.75 cde
8	nd a	106.95 ± 1.16 de	170.99 ± 0.91 bcde	817.54 ± 4.43 de	195.33 ± 2.43 bcd	411.60 ± 6.77 ab	157.84 ± 6.37 bc	14.73 ± 1.83 b	1874.99 ± 16.11 cd
9	nd a	nd a	150.04 ± 3.75 ab	770.46 ± 21.26 cd	173.91 ± 7.01 ab	378.76 ± 15.91 a	141.86 ± 2.71 ab	16.47 ± 0.11 bc	1631.49 ± 13.51 b
10	nd a	130.51 ± 10.12 f	185.91 ± 7.90 cde	846.72 ± 11.86 ef	212.52 ± 8.16 bcde	423.33 ± 3.80 ab	172.75 ± 1.80 cde	18.34 ± 2.96 bc	1990.08 ± 42.99 cde
11	nd a	94.95 ± 4.28 cde	141.79 ± 5.63 ab	821.37 ± 5.86 de	179.73 ± 11.86 abc	444.66 ± 14.36 bc	181.82 ± 1.81 de	nd a	1864.32 ± 11.47 c
12	nd a	81.01 ± 1.74 bc	165.77 ± 8.30 bcd	735.48 ± 8.30 bc	252.49 ± 1.05 e	681.13 ± 19.23 g	132.37 ± 7.25 a	22.61 ± 1.54 bcd	2070.84 ± 22.15 de
13	12.79 ± 0.49 b	95.84 ± 6.56 cde	202.52 ± 5.35 e	745.28 ± 24.41 bc	217.23 ± 18.37 cde	501.38 ± 8.38 def	191.63 ± 3.36 ef	30.10 ± 2.44 de	1983.99 ± 52.12 cde
Mean	3.85	89.35	166.28	776.77	199.84	472.23	170.75	17.82	1893.03
SD	7.84	31.33	24.53	107.54	27.32	82.03	26.14	10.17	187.74
CV (%)	203.59	35.07	14.75	13.85	13.67	17.37	15.31	57.06	9.92

a, b, …—means in the same column for all variants followed by different letters are significantly different (*p* ≤ 0.05). Values are mean ± SD (*n* = 3); SD—standard deviation; CV—coefficient of variance; nd—not detected.

**Table 3 molecules-29-01360-t003:** Tocol contents (mg/kg) in *N. sativa* L. seed oils.

Sample	α-Tocopherol	γ-Tocopherol	β-Tocotrienol	δ-Tocotrienol	Total
1	nd a	55.22 ± 1.57 bc	194.16 ± 2.35 cde	16.29 ± 9.87 bc	265.67 ± 10.65 b
2	20.14 ± 11.15 a	nd a	135.97 ± 3.18 a	nd a	156.12 ± 14.32 a
3	nd a	54.62 ± 13.11 b	197.79 ± 12.52 de	18.26 ± 2.71 bc	270.67 ± 22.92 b
4	tr a	83.77 ± 1.55 cd	207.84 ± 5.80 ef	25.59 ± 3.80 c	317.21 ± 8.05 bc
5	tr a	112.48 ± 15.23 e	240.68 ± 7.56 g	5.12 ± 7.24 ab	358.28 ± 15.55 c
6	nd a	nd a	140.01 ± 1.18 ab	nd a	140.01 ± 1.18 a
7	4.71 ± 1.61 a	nd a	137.79 ± 3.20 ab	nd a	142.50 ± 1.59 a
8	nd a	nd a	178.12 ± 2.49 cd	nd a	178.12 ± 2.49 a
9	nd a	nd a	182.31 ± 5.58 cde	nd a	182.31 ± 5.58 a
10	nd a	nd a	165.98 ± 13.58 bc	nd a	165.98 ± 13.58 a
11	tr a	nd a	147.82 ± 2.26 ab	nd a	147.82 ± 2.26 a
12	16.81 ± 18.05 a	360.35 ± 15.20 f	253.53 ± 10.69 g	nd a	630.69 ± 43.94 d
13	0.42 ± 0.59 a	91.72 ± 5.73 de	234.55 ± 8.50 fg	nd a	326.68 ± 13.64 bc
Mean	3.24	58.32	185.89	5.02	252.47
SD	6.92	99.90	40.28	8.91	137.65
CV (%)	213.76	171.29	21.67	177.46	54.52

a, b, …—means in the same column for all variants followed by different letters are significantly different (*p* ≤ 0.05). Values are mean ± SD (*n* = 3); SD—standard deviation; CV—coefficient of variance; nd—not detected; tr—traces.

**Table 4 molecules-29-01360-t004:** Quality features of *N. sativa* L. seed oils.

Sample	Conjugated Dienes (%)	Acid Value (mg/KOH/g)	Peroxide Value (meq O_2_/kg)
1	0.24 ± 0.00 c	28.2 ± 0.1 c	52.0 ± 0.6 c
2	0.17 ± 0.00 ab	52.3 ± 0.8 g	18.3 ± 0.3 ab
3	0.24 ± 0.01 c	30.0 ± 0.1 d	49.1 ± 0.2 c
4	0.22 ± 0.00 c	24.9 ± 0.7 b	46.3 ± 0.2 c
5	0.25 ± 0.00 c	19.7 ± 0.5 a	62.5 ± 0.2 d
6	0.18 ± 0.00 ab	37.6 ± 0.0 f	15.6 ± 0.7 a
7	0.16 ± 0.00 ab	34.9 ± 0.2 e	11.6 ± 0.1 a
8	0.17 ± 0.00 ab	36.6 ± 0.2 f	18.1 ± 0.4 ab
9	0.15 ± 0.00 a	28.3 ± 0.2 c	16.0 ± 0.1 ab
10	0.19 ± 0.01 b	23.6 ± 0.4 b	23.0 ± 0.2 b
11	0.18 ± 0.00 ab	24.7 ± 0.0 b	14.3 ± 0.1 a
12	0.25 ± 0.00 c	51.6 ± 0.3 g	71.6 ± 0.0 e
13	0.24 ± 0.02 c	37.5 ± 0.1 f	66.1 ± 6.2 de
Mean	0.20	33.08	35.73
SD	0.04	10.11	22.53
CV (%)	18.02	30.57	63.07

a, b, …—means in the same column for all variants followed by different letters are significantly different (*p* ≤ 0.05). Values are mean ± SD (*n* = 3); SD—standard deviation; CV—coefficient of variance.

## Data Availability

The raw data supporting the conclusions of this article will be made available by the authors on request.

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
