# Peer review of "Variation in the Composition and Quality of Nigella sativa L. Seed Oils—The Underestimated Impact on Possible Health-Promoting Properties"

_molecules, 2024, doi:10.3390/molecules29061360_

Round 1
Reviewer 1 Report
Comments and Suggestions for Authors
Thank you for the opportunity to evaluate this manuscript. I have found it interesting and significant, but it is unacceptable in its current form. Please address the following issues:
1. The authors stated that they used a BPX70 capillary column. According to the column's specification, it is suitable for fatty acid analysis and separation of cis and trans isomers. However, the authors did not report the content of isomers; they only provided the total content. Why so? The Isomers ratio might be interesting and significant. Analysis of the chromatograms should be repeated.
2. Please provide chromatograms of your samples in the Supplementary file. There is no figure of chromatogram in the manuscript. Please insert at least one chromatogram into the manuscript.
3. Please describe analytical methods and sample reparation protocols in detail.
4. How did you quantify phytosterols and squalene? Please describe the procedure.
5. Please present the results of the headspace analysis in the Table. The figure is poorly visible. Also, provide the chromatograms.
Generally, the discussion section should be expanded, not just to state presented results. Compare with the available results of the previously conducted studies.
Reviewer 2 Report
Comments and Suggestions for Authors
Comments for the authors of the article: Variation of the composition and quality of Nigella sativa L. 2 seed oils – the underestimated impact on possible health-promoting properties
According to the objectives indicated by the authors, the work aimed to verify the variability in composition and quality of Nigella sativa L. oil depending on the origin and expiration date of the Nigella sativa L. seeds.
In this work the authors describe the objective of the research work. They briefly describe the experimental design, the type of samples analyzed, the analytical methodologies applied and the results obtained. Finally, some ideas about this work conclude.
As stated by the argument presented by the authors, knowing the composition of Nigella sativa L. oils is important because there are studies on the biological, beneficial health effects of Nigella sativa L. oil, but, according to the authors, those studies do not characterize the oil they use.
According to the authors' arguments, the topic is of great interest since, probably by knowing the composition precisely and standardizing the characterization methods of said oil, one can make progress in choosing the best composition profile for the different biological purposes.
However, there are points for improvement that are listed below:
In the conclusions, the authors do not analyze or confirm whether the methodologies they used to characterize the oils can be considered a first step or a definitive step for the standardization of methodologies that allow these oils to be characterized. It is important to discuss it since it is the objective of the study and seems to be the main contribution of this research work. If they consider that other studies are required to standardize the methodology, it is necessary that they describe what studies are required to achieve the stated objective or to relate it to the biological effects mentioned in the bibliography.
On the other hand, the authors do not mention the relationship that could exist between the oils they analysed, or the composition found and the oils that have been reported in biological studies.
Line 162 mention that it is the first time that the presence of squalene has been reported. It would be desirable that:
· They will indicate whether this is desirable or not, for the biofunctional properties of Nigella sativa L. oil.
· They mention that the presence only in some samples may be due to the evolution in which the biosynthetic routes of this compound were found. Since this Journal specializes in molecular aspects, it would be important for them to discuss who the precursors are and perhaps report on possible biotic or abiotic factors that could promote the presence of this compound, whether because it is desirable or undesirable, this would shed light on the possible management of the seeds to have or not a certain squalene content.
187-189 Please review the sentence and rephrase, the idea is not very clear.
Method
Fatty acid, phytosterols and squalen conten
It is necessary to indicate how both fatty acids, phytosterols and squalen were quantified. If the authors used standards for this, it is important to decribe it and report the linearity of the calibration curves since an essential part of the objective is to verify the composition of the seeds, which requires indicators of reliability and repeatability.
Row 375-376: It is necessary to indicate the linearity of the calibration curves for the calculation of the tocol content.
Reviewer 3 Report
Comments and Suggestions for Authors
The article provides information in variation of the composition and quality of Nigella sativa L. seed oils, which is in recent years frequently tested for medicinal purposes. However, the information regarding quality control and chemical composition is still insignificant. The authors examined cumin seed oils from four countries for fatty acid composition, unsaponifiable fraction and volatiles. The overall scientific soundness of the paper and the English quality is good. Most of the cited references are from last 10 years and are properly selected and cited.
Major points
Line 264. Figure 1. Retention indices should be provided- the ones obtained by GC-MS as well as the ones from the literature
Lines 333/345/362/377. Representative chromatograms for all analysed compounds should be submitted in Supplementary files
Line 388. Can the authors provide a reference that justifies the narrow scan range m/z 10-200? The usual range for volatile compounds is up to 350-400.
Line 408. The authors claim that “The key accomplishment of this study was the finding of significant variations in N. sativa L. oils concerning their essential oils content among different samples...”, whilst the authors did not analyse the essential oil, but rather volatile composition of seed oil. This must be corrected.
Minor points
Line 88. The author discuses that pre-treatment and oil obtainment is significant for oil composition and quality, and that the cold pressing in contrast to Soxhlet and MAE results in lower induction. Is Soxhlet extraction of seed oil good method for human consumption due to the usage of organic solvent (n-hexane) during the extraction process? It is most certainly good for quality analysis, but can the authors provide a reference that justifies the use of Soxhlet in commercial production of seed oil?
Line 244. The authors should not address Figure 1. as to overall essential oil content since they did not analyse essential oil.
Line 264. Figure 1. The authors should add to the converted arbitrary units of peak area to % of peak area so it is easier to observe the differences of individual compounds as well
Line 271. Is there a particular component responsible for distinguishing Poland sample from others? Or was it just the overall amount of analysed phytochemicals?
Reviewer 4 Report
Comments and Suggestions for Authors
Dąbrowski et al. prepared n-hexan extracts of Nigella sativa seeds from different origins and evaluated the fatty acid composition, the unsaponifiable compounds content and the volatile compounds. In addition several quality tests of fatty acids were performed. The results obtained are discussed in the light of previous publications on Nigella sativa seed oils.
The authors have selected an interesting topic and performed the experimental part in a very good way. In addition the whole manuscript is written in a fluid style without grammar or spelling mistakes. Altogether 40 references are considered in this manuscript.
HOWEVER, there are some minor issues which should be reconsidered by the authors:
1. Line 16 (abstract) and in other parts of the manuscript, including the discussion and the conclusions: the authors mention that squalene was detected for the first time in samples of the Nigella sativa seed oil. However, another study by Javed et al., 2022, “Chemometrics-assisted Comparative Chemical Profiling of Marketed Nigella sativa L. Seed Oils Using Spectroscopic Techniques“, published in the Latin American Journal of Pharmacy 41 (10): 1917-29 (2022), should be considered. Please change your statement on squalene, e.g. mentioning that squalene was hitherto rarely identified in the seed oil of Nigella sativa.
2. LIne 291-308: In phytochemical studies voucher specimens should be stored. Please add information on this and also who performed a visual (or microscopical?) identity check of the seeds before starting the experiments.
3. Did you observe any differences relating to the colour or the size or weight of the seeds?
4. For the discussion of the results the authors should check again for different varieties of the title plant. Perhaps this whould also add to the differences observed in this study?
5. Another factor, the package of each sample, might have an influence on the essential oil contents. Please check this aspect again and add a remark in the discussion part if needed. I propose to add the kind of package of each sample in chapter 3.1.
6. Line 323: The quantities used for the solvent extraction procedure need to be added.
7. Lines 363.376: the temperature condition should be added for the HPLC-FLD study
8. In the reference list all plant names should be written in italics
To sum up, this is an interesting study which will be of great interest for the international readership of the journal. Only minor issues should be reconsidered by the authors.
Round 2
Reviewer 1 Report
Comments and Suggestions for Authors
Authors answered all issues and improved manuscript according to the reviewers suggestions. I recommend acceptance in its current form.
Author Response
We are glad that the corrections satisfied the Reviewer and thank for this positive comment.
Reviewer 3 Report
Comments and Suggestions for Authors
Major point
The authors have not properly changed Figure 1. RI should be added to Figure 1, and well as literature RI from NIST 2017 MS for the index’s comparison. Also, converted arbitrary units of peak area to % of peak area should be added to Figure 1.
Minor point
The authors have added revised version of Table 1 and now there are two rows (C18:0 & C18:1) with the same result for different compounds? Should be corrected.
Author Response
Reviewer comment 1:
The authors have not properly changed Figure 1. RI should be added to Figure 1, and well as literature RI from NIST 2017 MS for the index’s comparison. Also, converted arbitrary units of peak area to % of peak area should be added to Figure 1.
Authors answer:
The analyzed volatile compounds were identified only tentatively based on their mass spectra and matching to the NIST 2017 MS library considering over than 85% of match factor. We did not calculate the RI for them.
In the case of identified black cumin oil volatiles we presented their quantity as area units (AU/g of sample). The relative amounts of compounds in a sample can be determined by comparing the peak areas only for compounds with similar relative response factors (like eg. fatty acids in plant oil). Black cumin oil volatiles differ highly in molar mass and structure, so the same quantity gives different detector response. In such case, they shouldn’t be presented as percentage of summarized peaks area. Our study compares relative quantity of each volatile compound among used 13 samples. It points that for example thymoquinone content in sample No 13 was 190-fold higher than in sample No 7.
Reviewer comment 2:
The authors have added revised version of Table 1 and now there are two rows (C18:0 & C18:1) with the same result for different compounds? Should be corrected.
Authors answer:
We apologize for this error. A revised table is included in the current version of the manuscript.